# Transcriptomic Profiling of Young Cotyledons Response to Chilling Stress in Two Contrasting Cotton (*Gossypium hirsutum* L.) Genotypes at the Seedling Stage

**DOI:** 10.3390/ijms21145095

**Published:** 2020-07-19

**Authors:** Gongmin Cheng, Longyan Zhang, Hantao Wang, Jianhua Lu, Hengling Wei, Shuxun Yu

**Affiliations:** 1College of Agronomy, Northwest Agriculture and Forestry University, Yangling 712100, China; cgm900923@163.com; 2State Key Laboratory of Cotton Biology, Institute of Cotton Research, Chinese Academy of Agricultural Sciences, Anyang 455000, China; 15614991132@163.com (L.Z.); w.wanghantao@163.com (H.W.); lujh212760@163.com (J.L.); 3College of Agronomy, Hebei Agricultural University, Baoding 071001, China

**Keywords:** *Gossypium hirsutum*, young cotyledon, chilling stress, RNA-seq, WGCNA

## Abstract

Young cotyledons of cotton seedlings are most susceptible to chilling stress. To gain insight into the potential mechanism of cold tolerance of young cotton cotyledons, we conducted physiological and comparative transcriptome analysis of two varieties with contrasting phenotypes. The evaluation of chilling injury of young cotyledons among 74 cotton varieties revealed that H559 was the most tolerant and YM21 was the most sensitive. The physiological analysis found that the ROS scavenging ability was lower, and cell membrane damage was more severe in the cotyledons of YM21 than that of H559 under chilling stress. RNA-seq analysis identified a total of 44,998 expressed genes and 19,982 differentially expressed genes (DEGs) in young cotyledons of the two varieties under chilling stress. Weighted gene coexpression network analysis (WGCNA) of all DEGs revealed four significant modules with close correlation with specific samples. The GO-term enrichment analysis found that lots of genes in H559-specific modules were involved in plant resistance to abiotic stress. Kyoto Encyclopedia of Genes and Genomes (KEGG) pathway analysis revealed that pathways such as plant hormone signal transduction, MAPK signaling, and plant–pathogen interaction were related to chilling stress response. A total of 574 transcription factors and 936 hub genes in these modules were identified. Twenty hub genes were selected for qRT-PCR verification, revealing the reliability and accuracy of transcriptome data. These findings will lay a foundation for future research on the molecular mechanism of cold tolerance in cotyledons of cotton.

## 1. Introduction

Temperature is an essential environmental factor limiting the geographical distribution and growing season of plants [1]. Cold stress is classified as chilling (0–20 °C) and freezing (<0 °C), which are regulated by different mechanisms [2]. Low temperature is detrimental to plant growth and survival, as well as the yield and quality of thermophilic crops. As a commercial crop originating in low-latitude regions, the optimum growth temperature of cotton is 20–30 °C [3]. However, different varieties of the same plant species are generally with different tolerance to cold stress [4,5]. Tropical plants have been reported to be sensitive to cold stress, especially during the transition from heterotrophic to the autotrophic stage, and have different tolerance to cold stress at different growth stages [6,7]. Therefore, identification of tolerant and sensitive varieties at a specific growth stage is of great help to reveal the cold tolerance mechanism.

Physiological and biochemical changes in cold stress are often used as indexes to assess cold tolerance of plants. Antioxidant enzymes such as superoxide dismutase (SOD), peroxidase (POD), and catalase (CAT) help eliminate ROS accumulated excessively under abiotic stress to prevent plants from oxidative damage [8]. Activities of these antioxidant enzymes are often used as physiological indices for cold tolerance [9]. However, the severity of membrane damage and ROS burst levels reflect the degree of oxidative damage under stress and are usually used as negative indicators for evaluating cold tolerance [10,11]. Malondialdehyde (MDA) is the ultimate product of lipid peroxidation, and its dynamic accumulation in plant cells indicates the degree of membrane damage. ROS including hydrogen peroxide, hydroxyl radicals, and peroxides not only act as second messengers of cold stress, but their excessive accumulation in cells can lead to oxidative damage to plants. Additionally, soluble sugars and proline act as osmotic substances to protect plants from cold stress [12]. These biochemical indexes can help to understand the cold tolerance mechanism of plants.

In recent years, some cold tolerance mechanisms and genes have been discovered. For example, the ICE-CBF-COR transcriptional cascade is thought to be induced by cold stress and plays a vital role in cold response and adaptation of many plant species [13,14]. It is well known that CBFs, as AP2/ERF transcription factors, can bind to the CRT/DREB elements on promoters of cold-regulated (COR) genes to activate their expression. Overexpression of *AtCBF1* and *AtCBF3* in tomato, maize, tobacco, and rice enhanced cold tolerance [15,16,17,18]. Many cold-regulated genes have been reported to be related to plant cold tolerance. For instance, overexpression of *COR15a* in *Arabidopsis* could enhance the freezing resistance of chloroplasts and protoplasts [19]. Besides, some members of the ERF, bHLH, MYB, and C2H2 transcription factor families, such as *ERF105*, *ICE1*, *MYB4*, *MYB15*, *ZAT10*, and *ZAT12*, are also involved in regulating the expression of genes responding to cold stress [20,21,22]. Previous studies have shown that some phytohormones and their signal transduction-related genes are also crucial in response to cold stress [23]. However, the cold tolerance mechanism of cotton cotyledon remains unknown.

In plants, some candidate genes or loci have been identified by quantitative trait loci (QTL) mapping and genome-wide association analysis [24,25,26,27]. However, it is still challenging to mine cold resistance genes according to the genetic mapping of QTLs for some plant species. Fortunately, with the development of molecular biology techniques, RNA-seq is becoming a popular strategy to identify a large number of cold-responsive genes quickly and to elucidate the molecular mechanisms of cold tolerance in different plant species and organs at different development stages. In wheat, RNA-seq was used to study the cold response mechanisms of meristematic crown and leaves of spring and winter wheat seedlings, and flag leaves of durum wheat at the reproductive stage [28,29]. In rice, transcriptomic analysis has been used to study the cold response mechanisms of leaves, roots, and shoots at the seedling stage, as well as anthers at the reproductive stage [30,31,32]. Recently, scientists have studied the cold response mechanisms of true cotton leaves at the seedling stage and seed embryos at the germination stage by using RNA-seq technology [3,33]. However, reports on the transcriptome profiling of cotton seedling cotyledons under cold stress had not been disclosed.

Young cotyledons, as the first two opposite leaves, are an indispensable part of early cotton seedlings. They are not only the essential autotrophic organs before the first true leaf appeared but also the most vulnerable parts to cold stress. Previous studies on the response of cotton to cold stress mainly focused on the germination stage of seeds and the true leaf stage of seedlings [3,33]. However, the seedling cotyledons are more susceptible to cold stress than true leaves. Therefore, we focused our research on the cold tolerance of young cotyledons at the early seedling stage. In this study, to better understand the diversity in cotyledon cold tolerance of natural populations and to select valuable varieties, we first used the chilling injury (CI) index to evaluate the cold sensitivity of young cotyledons of 74 upland cotton varieties and selected two varieties (H559 and YM21) with contrasting phenotypes for subsequent studies. To comprehensively understand the molecular mechanism of cold tolerance in early seedlings, we performed physiological experiments and comparative transcriptome sequencing on the young cotyledons of both varieties. Through comparative transcriptome analysis, important expression patterns, significant pathways, and hub genes were identified, which would provide help for future research on the molecular mechanism of cotton cotyledon cold tolerance and breeding.

## 2. Results

### 2.1. Evaluation of Cold Tolerance Based on CI Index

We evaluated the chilling sensitivity of 74 upland cotton varieties from different ecological regions in China. The CI index of young cotyledons was adopted to measure the cold sensitivity of cotton seedlings. Seedlings on the seventh day after sowing were subjected to a chilling treatment regime of 36 h at 4 °C plus a week at 25 °C. After surveying the chilling injury phenotype and calculating the CI index of each variety, we found that there was a wide variety of cold tolerance among these varieties (Figure 1A). Of these varieties, YM21 appeared to be the most chilling-sensitive (CI index = 43.17%), and extensive cotyledon damage of most young seedlings was observed after seven-day recovery. In contrast, H559 was the most cold-tolerant (CI index = 4.47%), and its cotyledons showed minor damage after recovery (Figure 1A,B). Furthermore, the seedling survival rate of the two varieties after 48 h of cold stress (4 °C) was also investigated (Figure 1C). Two days of continuous chilling stress resulted in the death of a large number of YM21 seedlings (survival rate = 15.24%) while H559 was less affected (survival rate = 90.20%). Therefore, YM21 (chilling-sensitive) and H559 (chilling tolerant), two varieties with distinctively different chilling sensitivity, were selected for further analysis.

### 2.2. Physiological Responses to Chilling Stress in Cotton Cotyledons

To reveal the physiological response of young cotyledons of both varieties at different time points under cold stress, we determined the activities of SOD, POD, and CAT (Figure 2). When seedlings exposed to 4 °C for 24 h, the activities of the three enzymes in H559 increased at 3 h and then decreased, while the changing trend in YM21 was the opposite. In general, the enzyme activities in YM21 were higher than that of H559 during the cold stress process, except for CAT activity at 3 h. Obviously, there was a conflict between the results of enzyme activity tests and the CI evaluation.

To eliminate this misunderstanding, we measured the ROS (O^2−^ and H_2_O_2_) levels and ROS scavenging capability in cotyledons of the two varieties in the cold (Figure 2). With the extension of cold stress time, the level of O^2−^ increased slightly, but it was significantly lower in H559 than that of YM21. In the cotyledons of H559, the content of H_2_O_2_ decreased continuously and was always substantially lower than that of YM21. Compared with the control, H_2_O_2_ content in YM21 significantly increased when exposed to 4 °C for 24 h. In response to chilling stress, the superoxide anion scavenging (SAC) rate in YM21 was consistently lower than that in H559. These results suggest that H559 is more cold-tolerant in terms of ROS accumulation levels. MDA is the final product of membrane lipid peroxidation and is often used as a physiological index to measure the degree of cold damage. We found that the MDA level in H559 was significantly higher than that in YM21 at 3 h after cold stress, but it was the opposite at 24 h, indicating that the cold tolerance of H559 gradually increased along with the extension of cold stress (Figure 2).

Osmoregulation substances, including soluble sugar (SS), soluble protein (SP), and proline, protect plants from cold stress, and their concentrations were measured here (Figure 2). Under continuous cold stress, the levels of all osmotic adjustments in both varieties showed a rising trend. The content of SS and SP in YM21 was always higher than H559, while the proline content was lower than H559 after cold stress. These results suggest that the cold tolerance mechanism of young cotton cotyledon can be complicated, and changes of many physiological parameters in H559 are superior to YM21 in response to chilling stress.

### 2.3. Transcriptome Sequencing and Differential Expression Gene Analysis

To reveal the molecular mechanism of the difference in cold tolerance between YM21 and H559 at the early seedling stage, we performed transcriptome sequencing using total RNA extracted from the cotyledons at 0, 3, and 6 h of chilling stress. Eighteen cDNA libraries were constructed and sequenced using Illumina HiSeq2500 by Gene Denovo Biotechnology Co. (Guangzhou, China). Each cDNA library produced about 6.5 Gb to 8.9 Gb of raw data. All RNA-seq raw datasets were deposited in the NCBI database with an SRA accession number SUB7558830. After removing reads containing adapters and low-quality reads, the Q30 of each library ranged from 91.08% to 92.33%, and the GC content ranged from 45.83% to 46.78%. Approximately 92.52% to 97.47% clean reads in each library were mapped to the cotton genome (http://ibi.zju.edu.cn/cotton/), of which 87.6% to 92.27% were uniquely mapped. Besides, 4668 novel genes were identified (Appendix A). Details of transcriptome sequencing and alignment with the reference genome were shown in Table 1. The mapped reads of each sample were assembled by using StringTie v. 1.3.1 in a reference-based approach. For each transcription region, an FPKM (fragment per kilobase of transcript per million mapped reads) value was calculated to quantify its expression abundance and variations, using StringTie software. A total of 64,461 transcripts were obtained with FPKM values > 0 in at least one sample, of which 44,999 were expressed with FPKM values of ≥1. Therefore, we found that 44,998 genes were expressed in T0, and 344 of these genes were preferentially expressed in T0 (Figure 3A,C). Interestingly, the expressed genes in the T0 sample included all the expressed genes in the others, suggesting that differences in the genetic background might play an essential role in chilling tolerance of the two varieties.

Pearson correlation coefficient analysis based on the expression levels of all expressed genes showed that correlation coefficients between biological repeats ranged from 0.98 to 1.00, indicating that the sample repeatability was reliable (Appendix A). Cluster analysis showed that there was a close correlation between the precold stress and early response stages, and they were different from the samples at 24 h after cold stress. This indicates that the gene expression was less changed in the initial response stage of chilling stress.

Gene differential expression analysis was performed by using DESeq2 software between two different groups with the threshold of false discovery rate (FDR) < 0.05 and an absolute fold change of ≥2 [34]. In H559, 1626 DEGs were found in T0-vs-T3 (954 upregulated, 672 downregulated), 11,902 DEGs in T3-vs-T24 (4224 upregulated, 7678 downregulated), and 13,558 DEGs in T0-vs-T24 (4591 upregulated, 8967 downregulated; Table 2). In YM21, 2030 DEGs were found in S0-vs-S3 (1056 upregulated, 972 downregulated), 8984 DEGs in S3-vs-S24 (4152 upregulated, 4832 downregulated), and 11,795 DEGs in S0-vs-S24 (4522 upregulated, 7273 downregulated; Table 2).

### 2.4. Genetic Difference between Varieties at the Transcriptomic Level

Previous studies have reported that transcription differences among varieties before cold stress may be related to their genetic differences in response to cold stress [35]. Before cold stress, 44,999 genes expressed in H559 included all the genes (40,067) expressed in YM21. In addition, 1504 DEGs were detected between S0 and T0, of which 964 and 540 DEGs were upregulated in H559 and YM21, respectively (Figure 3B,C and Table 2). These findings suggested that H559 had a more complex background of transcriptional regulation to cope with cod stress than YM21. To further investigate the functional differences of these DEGs, GO-term enrichment analysis was performed (Appendix A). For the 964 DEGs upregulated in H559, 84 GO-terms were significantly enriched (*p* < 0.05) and the top three abundant terms in the biological process category were “single-organism metabolic process”, “response to stimulus”, and “carbohydrate metabolic process”. For the 540 DEGs upregulated in YM21, 77 GO-terms were significantly enriched (*p* < 0.05) and the terms “single-organism process”, “single-organism metabolic process”, and “single-organism cellular process” were the most abundant in the biological process. However, the GO-term “single-organism metabolic process” was the only one shared by both varieties. To remove the functional redundancy among GO-terms, we obtained overrepresented GO-terms by using the REVIGO program (http://revigo.irb.hr/). In H559, the top three significant clusters related to biological processes included “protein-DNA complex subunit organization”, “regulation of intracellular signal transduction”, and “malate transport” (Figure 4A). However, they were “pigment biosynthetic process”,“pigment metabolic process”, and “single-organism biosynthetic process” in YM21 (Figure 4B). These results suggest that the genes preferentially expressed in the two varieties have significant functional differences under suitable temperature, and H559 has a more sophisticated strategy than YM21 to deal with the possible sudden cold stress.

### 2.5. Weighted Gene Coexpression Network Analysis

To reveal the differences in gene regulation of cold response in cotton varieties with contrasting cold tolerance, weighted gene coexpression network analysis (WGCNA) was performed using 19,982 DEGs, and 11 merged coexpression gene modules were identified (Figure 5A). Module–sample relationship analysis found that four of the 11 modules had closer and significant correlations with particular samples (Figure 5C). Among them, the saddlebrown module contained 103 genes, which had a significant and positive correlation with T3 (*r* = 0.95, *p* = 2 × 10^−9^). Blue module containing 3743 genes had a closer correlation with T24 (*r* = 0.87, *p* = 2 × 10^−6^). This indicated that the genes of these two modules were related to the cold response of H559. In contrast, the paleturquoise and green modules had significant positive correlations with S3 (*r* = 0.86, *p* = 5 × 10^−6^) and S24 (*r* = 0.87, *p* = 2 × 10^−6^), respectively, suggesting that genes of these two modules were related to cold response of YM21. Additionally, a module–trait relationship analysis was performed using module eigengene and physiological data. As shown in Figure 5D, the saddlebrown and blue modules were highly positively correlated with MDA and proline, respectively. Previous studies have found that MDA mainly reflects the damage degree of cell membrane caused by abiotic stress, and an increase in proline content means enhanced cold resistance. The paleturquoise module was positively correlated with O^2−^, SS, and SP, but negatively correlated with SAC. It is known that level changes of O^2−^ (a kind of ROS) reflect the ability of cells to resist stress, while SS, SP, and SAC are positively correlated with cold tolerance of plants. There was a significant positive correlation between the green module and SP. Furthermore, we found that the darkmagenta module was highly positively correlated with SAC and that it was distinctively correlated with samples of contrasting genotypes. These results revealed the physiological and transcriptional differences between the cotyledons of H559 and YM21 in response to cold stress.

### 2.6. GO Enrichment Analysis and Kyoto Encyclopedia of Genes and Genomes (KEGG) Pathway Annotation

To reveal the functional differences of DEGs in the four significant modules, we performed GO-term enrichment analysis and genes in each module were recruited into three categories, including biological process, molecular functions, and cellular component (Appendix A). To get nonredundant GO-terms, de-redundancy analysis on the significant GO-terms (*p* < 0.01) was performed by using the REVIGO program [36]. Overall, 181 nonredundant and significant GO-terms were represented (Figure 6). In the blue module, “response to organonitrogen compound”, “defense response”, and “regulation of biological process” were the top three significant GO-terms for the biological process; “protein kinase activity”, “phosphotransferase activity”, and “kinase activity” were the top three significant GO-terms for the molecular function; whereas only “nuclear ubiquitin ligase complex” was significant for the cellular component. In the saddlebrown module, “glutamine family amino acid metabolic process”, “glutamine metabolic process”, and “response to stress” were the top three significant GO-terms for the biological process; “tetrapyrrole binding” was the most significant GO-term for the molecular function; whereas only “cell periphery” was significant for the cellular component. For the biological process, “response to stress” and “response to stimulus” were the common GO-terms enriched in both modules (blue and saddlebrown). Most terms in the two modules were related to plants resisting abiotic stress. In the green module, “regulation of gene expression”, “regulation of metabolic process”, and “regulation of biological process” were the top three significant GO-terms for the biological process; “nucleic acid binding transcription factor activity”, “DNA binding”, and “oxidoreductase activity” were the top three significant GO-terms for the molecular function; “membrane-bounded organelle” and “intracellular membrane-bounded organelle” were the most significant GO-terms for the cellular component. This suggested that transcription factors (TFs) played an essential role in response to chilling stress in cold-sensitive variety. In the paleturquoise module, “alpha-amino acid metabolic process”, “single-organism catabolic process”, and “CTP biosynthetic process” were the top three significant GO-terms for the biological process; “ligase activity”, “hydrolase activity”, and “anion binding” were the top three significant terms for the molecular function; whereas “cell wall” was the most significant for the cellular component.

To understand the function of genes in different modules more comprehensively, we also conducted KEGG pathway enrichment analysis. As shown in Table 3, a total of 24 significant pathways were obtained from the four significant modules. Among them, the plant hormone signal transduction pathway was shared by three modules of blue, green, and paleturquoise, and MAPK signaling pathway was shared by blue and green modules, indicating that the two pathways played a central role in the process of cotton cotyledon response and resistance to cold stress. In the blue module, the most abundant pathway was “plant–pathogen interaction”. In the saddlebrown module, “phenylpropanoid biosynthesis” and “glutathione metabolism” pathways were specially and significantly enriched. These results indicated that there were both common and unique pathways in cotyledons of the two cotton varieties in response to cold stress.

### 2.7. Construction of Gene Coexpression Network and Identification of Centrally Connected Genes

Hub genes usually have high connectivity in coexpression networks as indicated by high Kme value. According to the gene expression heatmaps and eigengene histograms of the four significant modules, we discovered that genes in the same module generally had the same preponderant expression stage and that genes in different modules often had different preponderant expression stages (Figure 7A, Figure 8A, Appendix A). The predominantly expressed genes at every time point can be assessed by stage specificity score (SSC) [37]. To mine highly expressed genes in preponderant expression stages, we normalized the expression levels of all genes within the same stage by using the Z-score method. The genes with higher Z-score values were considered to have higher expression levels at this specific stage. In our study, genes with SSC > 0.5 and Kme > 0.8 were defined as hub genes (Figure 7B, Figure 8B, Appendix A). A total of 662, 186, 39, and 49 hub genes were identified in blue, green, paleturquoise, and saddlebrown modules, respectively. The top five hub genes with high Z-score values in the preponderant expression stage of each module were shown in Table 4.

In the blue module, the top 100 hub genes with higher Z-score values at T24 and weight values >0.45 were used to construct the coexpression network. As shown in Figure 7C, 13 transcription factors were identified, of which *ERF4*, *ERF026*, *ERF017,* and *SCL13* were the critical transcription factor genes. Among them, *ERF4* (ethylene-responsive element-binding factor 4) was one of the top three hub genes with a higher degree. In the coexpression network of the saddlebrown module, the gene with the highest degree was *BH0283* (46 edges), followed by *BETV1F* (45 edges; Figure 8C). Key genes in the coexpression network of the green module included *GH_A10G0211* (hypothetical protein CCACVL1_15597), *GH_A05G4197* (sigma factor binding protein 2) and *GH_D08G1892* (probable serine/threonine-protein kinase PBL19; Appendix A). In the coexpression network of paleturquoise module, hub genes were almost correlated with each other, of which *TIP1-1* (aquaporin TIP1-1-like) and *BZIP1* (basic leucine zipper 1-like) were identified as key genes (Appendix A).

### 2.8. Transcription Factors in Response to Chilling Stress

Transcription factors (TFs) play an essential role in plant response to abiotic stress [38]. GO-term analysis of the genes in the blue and green modules revealed that quite a number of them were enriched to “regulation of biological process”, suggesting that TFs participated in regulating cold response. To better understand the transcription differences of TF genes in different genotypes and response stages, we performed TF analysis on the significant modules. A total of 594 TFs were identified, including 401 in the blue module, 155 in the green module, 35 in the paleturquoise module, and two in the saddlebrown module (Appendix A). This indicated that more TFs were involved in regulating cold response after exposure to cold stress for 24 h compared with 3 h. In the blue module, the top three abundant TF families were ERF (70 genes, 17.46%), WRKY (41 genes, 10.22%), and MYB (39 genes, 9.73%), while the key TF genes include *ZAT10* (*GH_D05G2128*), *RAV1* (*GH_A03G1130*), and *ERF4* (*GH_D01G0569*; Figure 9 and Appendix A). In the green module, the three major TF families were ERF (41 genes, 26.45%), WRKY (16 genes, 10.32%), and NAC (15 genes, 9.68%), and the key TF genes included *ERF5* (*GH_A12G2350*) and *ERF1B* (*GH_D02G0425*; Figure 8 and Appendix A). There was only one hub TF gene *BZIP1* (*GH_D09G0794*) in the paleturquoise module, but none had been identified in the saddlebrown module.

### 2.9. Genes Related to Plant Hormone Signal Transduction

Phytohormones play an important role in the response of plants to various abiotic stresses [39]. In our study, 133 genes from blue, green, and paleturquoise modules were significantly enriched in the “plant hormone signal transduction pathway”, and three genes from the saddlebrown module were also detected in this pathway (Figure 10). The differences in expression patterns and expression levels of phytohormone-related genes were shown in Figure 10B. In the auxin (IAA) signal transduction pathway, a total of 46 genes were detected, including two *AUX1* genes, 12 *AUX*/*IAA* genes, one *ARF* gene, 13 *GH3* genes, and 18 *SAUR* genes, of which 30 genes were from the blue module. After 24 h of cold stress, most *IAA* genes were significantly upregulated, especially those in the blue module showed higher expression levels, indicating that the *IAA* genes might have an important role in regulating the cold response of cotyledons. A total of 20 genes were found in the abscisic acid (ABA) pathway, including two *ABF* genes, six *PP2C* genes, eight *PYL* genes, and four *SnPK2* genes. Moreover, a total of 22 genes were found in the brassinosteroid (BR) pathway, 20 of which were from the blue module, including three *BAK1* genes, one *BRI1* gene, one *BSK* gene, one *BIN2* gene, two *CYCD3* genes, and 12 *TCH4* genes. These *TCH4* genes were all from blue and green modules and had higher expression levels, indicating that they mainly played a role in cotyledon cold adaptation and tolerance. Furthermore, 22 genes were also found in the ethylene (ET) pathway, including three *ETR* genes, one *SIMKK* gene (*MKK4*/*5*), one *EIN2* gene, three *EIN3* genes, five *EBF1*/*2* genes, and nine *ERF1* genes. In addition, 12 genes were identified in the jasmonic acid (JA) pathway, including one *JAR1* gene, four *JAZ* genes, and seven *MYC2* genes. We also found two *GID1* genes, three *DELLA* genes and three *PIF* genes (TF) in the gibberellin (GA) pathway, and one *NPR1* gene, three *TGA* genes, and one *PR1* gene in salicylic acid (SA) pathway. Remarkably, all three genes in the cytokinin (CK) pathway were from the blue module, including one *AHP* gene and two *ARR-A* genes.

### 2.10. Verification of RNA-Seq Data by qRT-PCR

To validate the accuracy and reliability of RNA-seq data and obtain candidate genes for further functional studies, we performed quantitative real-time PCR (qRT-PCR) for 20 interesting hub genes selected from four significant WGCNA modules. The 20 genes and their qRT-PCR primers were listed in Appendix A. We found that the relative expression levels of genes obtained by qRT-PCR were highly consistent with the RNA-seq data (Figure 11 and Appendix A). Linear regression analysis showed that the correlation between RNA-seq and qRT-PCR data was significantly positive (Figure 11B,C). These results suggested that the RNA-seq data was credible and accurate.

## 3. Discussion

Cotton (*Gossypium hirsutum* L.) is sensitive to chilling stress, especially at seedling emergence and early cotyledon stage. We had previously found that under low natural temperature, the germination and cotyledon stages of cotton would last for a long time. Relevant evidence also supported that low temperatures could reduce germination speed and delay plant growth [40]. Once the ambient temperature is suitable, cotton seedlings will quickly resume growth and development. Therefore, improving cold tolerance of young cotyledons is more meaningful for cotton seedlings. Some researchers have noticed this and assessed the cold sensitivity of young cotyledons of different cotton varieties with the help of CI index [41]. We found that cold stress (4 °C) for 24–36 h could not cause significant damage to roots and stems of most cotton seedlings, but would cause varying injury degrees of cotyledons. We used the revised CI index to evaluate the cold sensitivity of 74 cotton varieties, and found that H559 was the most tolerant and YM21 was the most sensitive. To elucidate the molecular mechanism of cotton cotyledons response and resistant to cold, physiological changes and transcriptomic analysis was performed in our study.

### 3.1. Physiological Response of Cotton Cotyledons to Chilling Stress

A plant will face various temperature stresses during its lifetime, including freezing, chilling, and high temperature. However, molecular mechanisms of plant responses and resistance to different temperature stresses are usually different [42]. To cope with cold stress, plants have evolved many strategies to help them survive and maintain growth, such as keeping ROS balance and accumulating osmotic substances [43,44]. In plants, ROS include hydrogen peroxide (H_2_O_2_), singlet oxygen, and superoxide anions, and their levels in cells are usually regulated by various antioxidant enzymes, such as CAT, POD, and SOD [45]. MDA is the ultimate product of phospholipid peroxidation, and its content reflects the integrity of cell membranes [46]. Under cold stress, MDA content increased rapidly and then decreased to normal levels in H559, indicating that cold-tolerant variety could quickly start the cold defense system to maintain and enhance the integrity of the cell membrane. Besides, the contents of H_2_O_2_ and O^2−^ reflect the ROS levels in cells, and their excessive accumulation will disrupt intracellular ROS balance and cause oxidative damage [47]. In our study, the contents of O^2−^ and H_2_O_2_ in H559 were significantly lower than YM21 under both normal and low temperatures, and the H_2_O_2_ content in H559 showed a reduction trend, indicating that the tolerant variety could restrict the accumulation of ROS in the cold. Under abiotic stress, various osmoregulatory substances such as soluble sugars (SS) and soluble proteins (SP) accumulate rapidly to enhance plant stress resistance [44,48]. Before and within 24 h of cold stress, the intracellular SS and SP concentrations of cold-sensitive variety were consistently and significantly higher than that of tolerant variety, which was similar to previous studies [41,49]. This may indicate that cotyledons of tolerant variety are not highly dependent on osmotic substances to improve cold tolerance in the early stage of cold response. It is well known that the activity of antioxidant enzymes represents ROS removal capability [8]. Compared with YM21, antioxidant enzyme activities increased rapidly in H559 in cold stress to remove over accumulated ROS. Therefore, we believe that these results of enzyme (SOD and POD) activities, which are different from previous research [41], are mainly due to the different mechanisms of cold resistance between cotyledons and true leaves. The cold tolerance of cotton cotyledons may be more dependent on genetic background and transcriptional regulation of the varieties.

### 3.2. Transcriptional Differences in Cotyledons of Varieties with Different Cold Sensitivity

Transcriptome technology has been used to reveal differences in the molecular mechanisms of cold tolerance among different varieties [33,50,51,52,53]. Previous studies have shown that there are more constitutive expression genes in the tolerant variety than the sensitive one before cold stress treatment [54,55]. It is believed that there should be relatively more genes involved in forming cold-tolerant genotypes. Under normal temperature, 40,067 and 44,999 expressed genes were detected in young cotyledons of YM21 and H559, respectively. It was noteworthy that expressed genes in T0 cotyledons included all the expressed genes of S0 and all the DEGs obtained by comparative analysis. GO enrichment analysis of the up and downregulated expressed genes acquired by comparing S0 with T0 revealed that their biological functions were significantly different, which was consistent with previous findings [5]. Thus, the constitutive diversities between varieties can result in different cold tolerance. In this study, WGCNA based on all DEGs helped us get four significant modules: early response module of YM21 (paleturquoise), late response module of YM21 (green), early response module of H559 (saddlebrown) and late response module of H559 (blue). In terms of the number of genes contained in each module, the paleturquoise module has more genes than the saddlebrown module in the early response stage, while the green module has fewer genes than the blue module in late response stage. Additionally, we found that the saddlebrown module was significantly correlated with MDA, while the paleturquoise module was significantly correlated with SS and SP. These results indicate that the early cold response of H559 does not perform as well as that of YM21. Many previous studies on plant cold tolerance have found that genes related to plant hormone signal transduction and protein kinases are induced by cold stress [56,57,58]. KEGG pathway analysis found that plant hormone signal transduction and MAPK signal pathways were significantly enriched in both tolerant and sensitive varieties at 24 h of cold stress, suggesting that they played important roles in regulating cold tolerance of cotton cotyledons.

### 3.3. Genes Related to Phytohormone Signal Transduction Involved in Cotyledon Cold Response

Phytohormones mainly include IAA, GA, CK, ABA, ET, BR, SA, JA, and strigolactone, which are small endogenous signaling molecules. Among them, BR, ABA, IAA, ET, GA, and JA have been reported to respond to cold stress and regulate cold resistance [59,60,61]. Previous studies have demonstrated that JA positively regulated plant cold tolerance [62]. In *Arabidopsis thaliana*, the low temperature can elevate the level of endogenous JA by inducing the expression of JA biosynthetic genes [63]. Moreover, the external application of JA improves cold tolerance in *Arabidopsis*. In this study, we found that some genes related to JA biosynthesis, including *LOX*s, *AOS,* and *AOC*, and genes associated with JA signal transduction including *JAR1*, *JAZ,* and *MYC2* were predominantly expressed in T24. This means that the JA signal plays an important role in regulating cotton cotyledon cold tolerance. It is known that BR treatment can improve the cold resistance of plants [64,65]. It was reported that overexpression of *BRI1* improved cold tolerance in *Arabidopsis*, whereas the *bri1* mutants showed cold hypersensitivity [66]. Some studies have suggested that *TCH4*, as a member of the BR signal transduction pathway, can respond to auxin and BR, as well as various environmental stimuli, including cold and heat shock [67,68]. In the present study, we detected one *BRI1* gene and 12 *TCH4* genes predominantly expressed in T24 and found that most genes in the BR signaling pathway were exclusively present in the blue module, indicating that BR signaling was necessary for cotyledon cold tolerance. Additionally, some studies have demonstrated that genes involved in the auxin signaling pathway are differentially expressed when plants are subjected to cold stress [30,69]. In these remarkable modules, we identified 136 genes involved in plant hormone signaling pathway, of which 46 were related to auxin signal transduction, most of which were from the blue module, indicating that auxin played a significant role in regulating cotyledon cold tolerance.

### 3.4. Protein Kinase-Related Genes Involved in Cotyledon Cold Response

Plant MAPK cascade participates in signal transduction induced by abiotic stress [70,71,72]. In *Arabidopsis*, ROS activate MAPK cascades to mediate the transmission of cold signals [73]. This indicates that the MAPK signal transduction pathway is important for plant response to cold stress. Our results demonstrated that the MAPK signaling pathway was significantly enriched in blue and green modules. The MAPK cascade transmits and amplifies signals by phosphorylating target proteins through three phosphorylation kinases (MPKs, MEKs, and MEKKs), and those phosphorylated proteins will alter the expression of downstream genes, including transcription factors and COR genes [26,71,72]. Some studies have shown that the MAPK pathway mediated by MEKK1-MKK2-MPK4/6 positively responds to cold stress [73,74]. In *Arabidopsis*, MKK2 can be induced by cold stress, and plants overexpressing MKK2 exhibit strong cold tolerance [73]. It is reported that overexpression of *OsMKK6* activates *OsMPK3* and enhances the cold endurance of rice [75]. However, the single *mpk3* mutant of rice showed freezing resistance [76]. These two diametrically opposite results are presumed to be caused by different degrees of low temperature. In our study, both *MKK2* and *MPK6* were upregulated under chilling stress. Additionally, we also identified that *MPK4*, *MKK4*/*5*, *MKK3*, *MKK9,* and *MEKK17*/*18* were highly expressed when H559 was subjected to cold stress, suggesting that these genes might be involved in regulating cold tolerance of H559 cotyledons. It is known that CDPKs and CIPKs are two protein kinases involved in the calcium signal transduction pathway and participate in plant response to cold stress [77,78]. Under low temperature, the transcription of many *CDPK* and *CIPK* genes in plants were induced [52,79]. It has been reported that some *CIPK* and *CDPK* genes play important roles in improving the cold resistance of plants [80,81,82,83,84,85]. In this study, 18 *CIPK* genes and 14 *CPK* genes were detected in blue (15 *CIPK* and 13 *CPK* genes) and green (three *CIPK* genes and one *CPK* gene) modules, indicating that most *CPK* and *CIPK* genes were preferentially expressed in the cold-tolerant variety. These genes will provide important implications for further research on the cold tolerance of cotton cotyledons.

### 3.5. Major TFs Involved in the Cold Response of Cotton Cotyledons

TFs play an essential role in regulating plant growth and development, as well as during abiotic stress response. When exposed to a low-temperature environment, TFs induced by cold stress are activated by various signaling pathways. The activated TFs can specifically bind to corresponding cis-acting elements of their target gene promoters to activate cold-responsive genes, thereby regulating the cold tolerance of plants [86]. In the present study, the AP2/EFR family contained most members, followed by WRKY and MYB families, suggesting that they played an important role in the response of cotton cotyledons to chilling stress. DREB is a subfamily of ERF TF family, and its members *DREB1B*/*CBF1*, *DREB1C*/*CBF2,* and *DREB1A*/*CBF3* are induced by cold stress [87]. Overexpression of *DREB*s/*CBF*s in *Arabidopsis* improved cold tolerance of plants, while suppression of *DREB1B* or *DREB1A* reduced plant cold tolerance [88,89,90]. In our study, two *DREB1B*/*CBF1* genes (*GH_A12G2453* and *GH_D12G2465*) were identified, of which *GH_A12G2453* had higher expression levels induced by chilling stress in H559. Similarly, some members of WRKY and MYB families, such as *VaWRKY12*, *VbWRKY32*, *AtMYB14*, and *MdMYB23* have been reported to participate in regulating cold tolerance of plants [91,92,93,94]. In addition, some other TFs families were upregulated in the cold-tolerant variety, such as C2H2, NAC, GRAS, bHLH, and bZIP. It is reported that *ZAT6*, *ZAT10,* and *ZAT12*, as members of the C2H2 family, participate in cold response [95]. In our study, a *ZAT10* gene was identified as a hub gene in the blue module, with higher expression in H559. Excitingly, most of the differentially expressed TF genes in the tolerant variety were involved in response to cold stress, and they would provide promising prospects for the improvement of cold tolerance in cotton cotyledons.

## 4. Materials and Methods 

### 4.1. Plant Materials and Cold Treatments

Seventy-four upland cotton varieties (provided by the Institute of Cotton Research of CAAS) were used to evaluate cotyledon cold tolerance. When the seedlings grew to the 6th day after sowing, plants with deformed cotyledons or weak growth were pulled out. Three biological replicates were set for each variety, and 50 to 60 seedlings were retained for each replicate. On the seventh day, plug seedlings were transferred to cold light plant growth chamber (15 h light/9 h dark), and after 36 h of cold stress (4 °C), they were returned to the cotton cultured room (15 h light/9 h dark, 25 °C) for another seven days. After a week of recovery, the chilling injury phenotype was investigated, and cold sensitivity was evaluated with the CI index. The cold-tolerant variety H559 and sensitive variety YM21 (also known as Yumian21) were selected from the 74 varieties. The seventh-day seedlings of both varieties were subjected to 4 °C for 0 to 24 h. Three replicates were set for each variety at each time point, and six cotyledons (from 6 seedlings) were collected for each replicate. T0, T3, and T24 represent cotyledon samples of H559 under cold stress at 0 h, 3 h, and 24 h, respectively. Similarly, S0, S3, and S24 represent cotyledon samples of YM21 under cold stress at 0 h, 3 h, and 24 h, respectively. The collected samples (T0 to T24 and S0 to S24) were used for physiological indices determination and RNA-seq. At the same time, seedlings of the two varieties were cold-treated for 48 h to investigate the survival rate after 14 days of recovery, and the emergence of true leaves was used as the criterion for seedling survival.

### 4.2. CI Index Evaluation and Investigation of Seedling Survival Rate

To evaluate the CI index of different varieties, we first graded cold-damage symptoms of single cotyledon according to the necrotic area (*X*): 0 (healthy), 1 (yellowish), 2 (*X* < 1/8), 3 (1/8 ≤ *X* < 1/4), 4 (1/4 ≤ *X* < 1/2), 5 (1/2 ≤ *X* < 3/4), and 6 (*X* ≥ 3/4). Then, the cold-damage symptoms of the cotyledons of individual seedling were graded according to the necrotic area (*Y*): 0 (*Y* = 0), 1 (*Y* < 1/16), 2 (1/16 < *Y* ≤ 1/8), 3 (1/8 < *Y* ≤ 1/4), 4 (1/4 < *Y* ≤ 3/8), 5 (3/8 < *Y* ≤ 1/2), 6 (1/2 < *Y* ≤ 3/4), and 7 (*Y* > 3/4). The CI index for a specific variety is denoted as: *CI* = 100 × (0 × *S_0_* + 1 × *S_1_* + 2 × *S_2_* + 3 × *S_3_* + 4 × *S_4_* + 5 × *S_5_* + 6 × *S_6_* + 7 × *S_7_*)/8*N*, where *S_i_* represents the number of seedlings at grade *i*, and *N* represents the total number of seedlings. Therefore, the CI value ranged from 0 to 1, and the lower the CI value, the more cold-resistant the variety. The CI evaluation method used here was an improvement and optimization based on a previous study [41].

### 4.3. Determination of Physiological Indices

Young cotyledons were sampled at 0, 3, and 24 h after cold stress and frozen quickly in liquid nitrogen. All samples were prepared for three biological replicates. All physiological indices, including superoxide dismutase (SOD) activity, peroxidase (POD) activity, catalase (CAT) activity, malondialdehyde (MDA) content, soluble sugar (SS) content, soluble protein (SP) content, superoxide anion (O^2−^) content, H_2_O_2_ content, proline content, and superoxide anion scavenging (SAC) rate, were determined using assay kits (Comin, Suzhou, China). The SPSS statistical software (SPSS version 19.0) was used for the analysis of variance.

### 4.4. RNA Extraction, Transcriptome Sequencing, and Data Analysis

Young cotyledons of H559 and YM21 seedlings were collected after 0, 3, and 24 h of cold stress and quickly frozen in liquid nitrogen. Each sample included three biological replicates, and each replicate consisted of six young cotyledons collected from six seedlings. Total RNA for each sample was extracted according to the Trizol reagent kit (Invitrogen, Carlsbad, CA, USA) guidelines. RNA quality was examined with an Agilent 2100 Bioanalyzer (Agilent Technologies, Palo Alto, CA, USA) and checked using RNase free agarose gel electrophoresis. Preparation of 18 cDNA libraries and transcriptome sequencing using the Illumina HiSeq2500 platform were completed by Gene Denovo Biotechnology Co. (Guangzhou, China). After raw reads were filtered, the obtained clean reads were aligned to the latest version of the Gossypium hirsutum genome, which can be available at http://ibi.zju.edu.cn/cotton. The FPKM value was calculated using StringTie v. 1.3.1 software to assess the abundance of each transcript [96]. Pearson correlation coefficient analysis was performed with all FPKM > 0 genes. The DESeq2 software was used to analyze the DEGs, and the genes with the parameter of FDR < 0.05 and an absolute fold change of ≥2 were considered differentially expressed [34]. For the specific comparison of DEGs, GO enrichment, and REVIGO (http://revigo.irb.hr) deredundancy analysis were performed [36].

### 4.5. Weighted Gene Coexpression Network Analysis

A total of 19,982 DEGs were used for WGCNA analysis. Coexpression modules were discovered using WGCNA (v. 1.47) package in R software with default settings, except that the power was 12, mergeCutHeight was 0.75, and minModuleSize was 50. To find out biologically significant modules, we used module eigengenes to calculate correlation coefficients with samples or physiological traits. Intramodular connectivity and stage specificity score (SSC) of each gene were calculated, and genes with higher SSC and connectivity tended to be hub genes. The SSC score was calculated as previously reported by Zhan et al. [37]. The networks were visualized using Gephi 0.9.2 [97]. For genes in each module, GO-term and KEGG pathway enrichment analysis were conducted to reveal the biological functions of modules. The REVIGO program (http://revigo.irb.hr/) was used to remove redundant GO-terms, and heatmaps were used to visualize nonredundant GO-terms.

### 4.6. RNA-Seq Data Validation

Young cotyledons were sampled at 0, 3, 6, 12, and 24 h after cold stress and frozen quickly in liquid nitrogen. The total RNA of each sample was extracted according to the instructions of RNAprep Pure Plant Plus Kit (TIANGEN Biotech Co.,Ltd., Beijing, China). All qRT-PCR primers were designed using Oligo7 software and synthesized by Sangon Biotech Co., Ltd. (Shanghai, China). The *GhACTIN7* (AY305733) gene was used as an internal reference gene [98]. According to the system and procedures recommended by the UltraSYBR Mixture (Beijing ComWin Biotech Co.,Ltd., Beijing, China) instructions, qRT-PCR was performed on a 7500 real-time PCR system. The relative expression of genes was analyzed by the 2^−ΔΔCt^ method. The verification experiment was performed in three biological replicates, with three technical replicates in each replicate.

## 5. Conclusions

In this study, we successfully identified outstanding cold-tolerant varieties through CI evaluation of young cotyledons after chilling stress and found that ROS levels and ROS scavenging capability were reliable physiological indices that could best reflect the cold resistance of cotyledons. Transcriptome data revealed that tolerant cotyledons had more sophisticated cold-induced transcriptional regulation capabilities than sensitive cotyledons before and after cold stress. WGCNA provided us with gene expression modules (blue, green, saddlebrown, and paleturquoise) and functional pathways (plant hormone signal transduction, MAPK signal, and plant–pathogen interaction) that positively respond to cold stress. These findings could not only help us understand the potential molecular mechanisms of cold tolerance in cotton cotyledons, but also laid a foundation for future research on the roles of identified hub genes in the regulation network of cold tolerance and breeding of cold-resistant varieties.

## Figures and Tables

**Figure 1 ijms-21-05095-f001:**
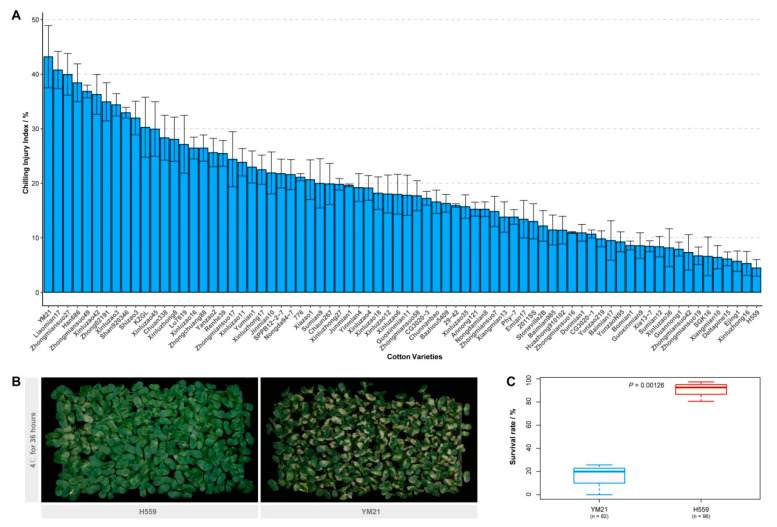
Assessment of seedling chilling injury (CI) index and investigation of survival rate. (**A**) Evaluation of the cotyledon CI index of 74 upland cotton varieties. The seedlings growing to the seventh day were treated with cold stress (4 °C) for 36 h and then recovered for another week, after which the chilling phenotype was investigated. Error bar indicates the standard error (SE) of the mean (*n* = 3). (**B**) Chilling injury symptoms of H559 and YM21 seedlings recovered one week after 4 °C stress. (**C**) The seedling survival rate of H559 and YM21 after 48 h of chilling stress.

**Figure 2 ijms-21-05095-f002:**
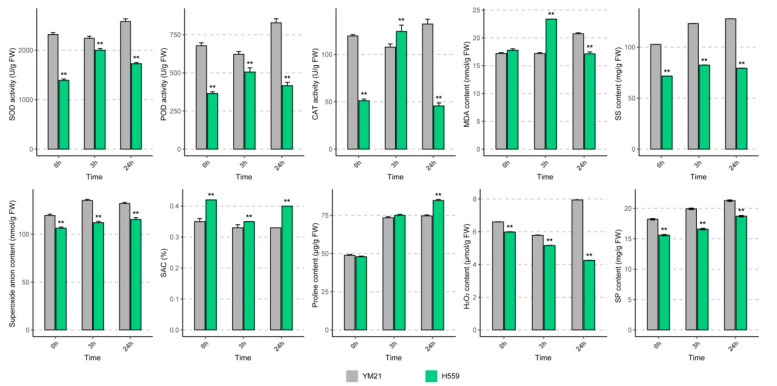
Physiological changes in young cotyledons of H559 and YM21 under chilling stress (4 °C) at different time points. These physiological indices include superoxide dismutase (SOD) activity, peroxidase (POD) activity, catalase (CAT) activity, malondialdehyde (MDA) content, soluble sugar (SS) content, soluble protein (SP) content, O^2−^ content, H_2_O_2_ content, proline content, and superoxide anion scavenging (SAC) rate. Error bar indicates the standard error (SE) of the mean (*n* = 3). The asterisk indicates a significant difference level between H559 and YM21 (** *p* < 0.01).

**Figure 3 ijms-21-05095-f003:**
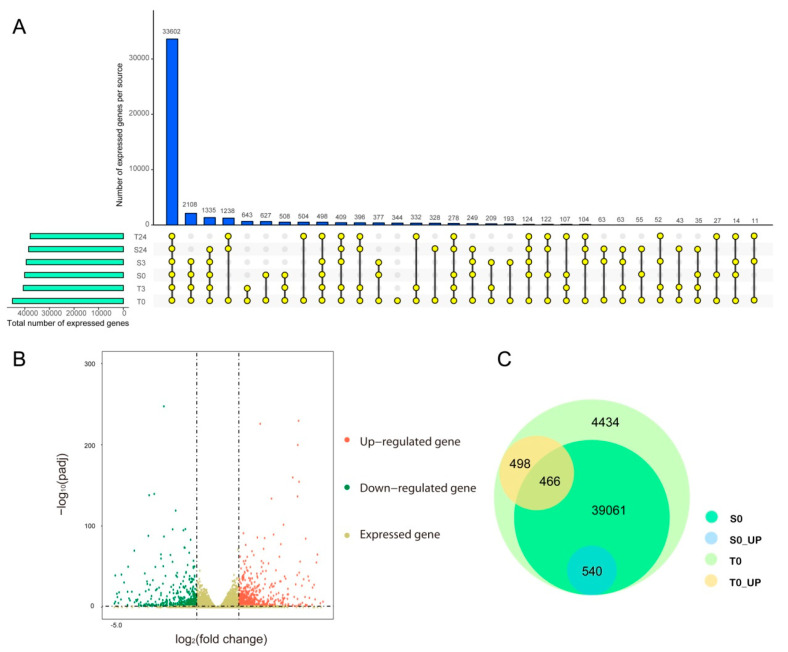
Venn diagram analysis of expressed genes in all samples and analysis of constitutive transcriptional differences between H559 and YM21. (**A**) The number of expressed genes identified in different samples. The horizontal bar shows the total number of expressed genes identified in each sample. The colored dots connected by lines indicate the sharing of genes between samples, and the number of genes corresponds to the vertical bar above. (**B**) The upregulated or downregulated genes in S0 compared to T0. The colored dots represent genes, where the red dots represent upregulated genes, the green dots represent downregulated genes in T0, and the khaki dots only represent expressed genes. (**C**) Venn diagram of expressed and differentially expressed genes within and between varieties before cold stress.

**Figure 4 ijms-21-05095-f004:**
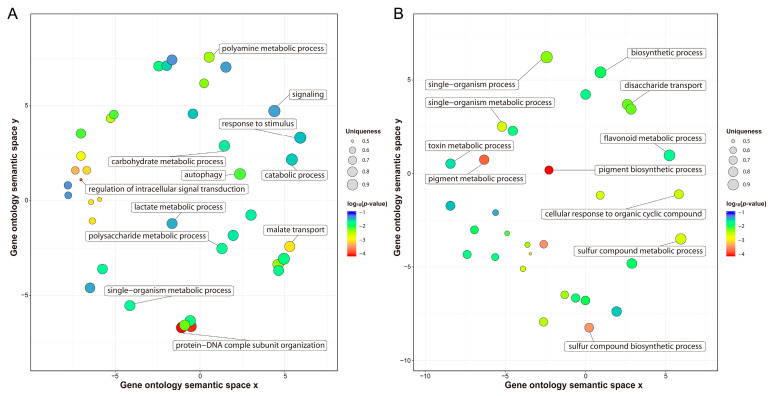
Nonredundant GO-terms related to biological processes obtained by enrichment analysis of the upregulated genes in H559 (**A**) and YM21 (**B**) before cold stress. The REVIGO program was used to de-redundant the significant GO-terms, and bubble charts were then drawn by using the R package to display the nonredundant GO-terms. Nonredundant GO-terms were clustered on a two-dimensional space according to the semantic similarity between them (adjacent bubbles are most closely connected). Bubble size is proportional to the uniqueness value of each GO-term, and the color represents the log_10_ (*p*-value).

**Figure 5 ijms-21-05095-f005:**
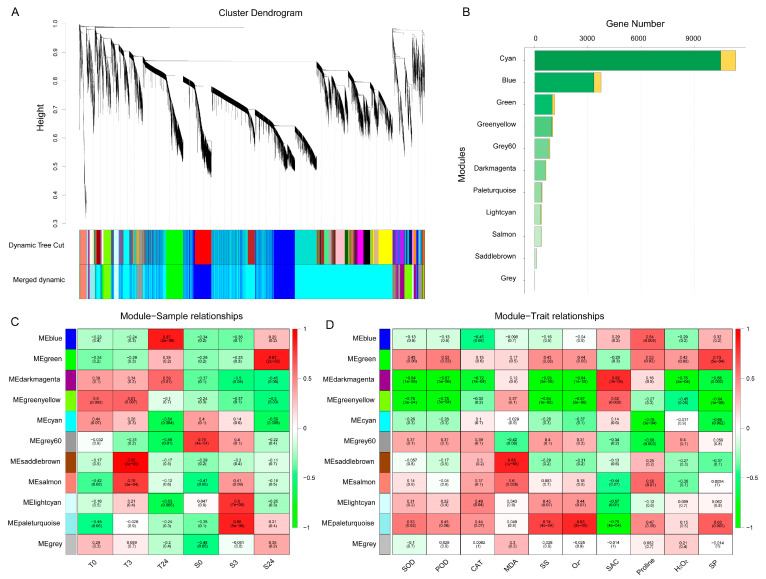
Weighted gene coexpression network analysis (WGCNA) of 19,982 DEGs obtained from all pairwise comparisons. (**A**) Hierarchical clustering tree showing 11 coexpression modules identified by WGCNA. Different modules are marked with different colors. Each leaf of the cluster tree represents a gene. (**B**) The number of transcription factors and nontranscription factor genes for each module. The orange bar indicates transcription factors, and the green bar indicates nontranscription factor genes. (**C**) Module–sample correlations and corresponding *p*-values. Each row represents a specific module, and each column represents a sample. The heatmap on the right shows the Pearson correlation between module eigengenes and samples. The numbers in each cell represent the correlation coefficients and correlation significance levels (in parentheses). The color of the cell reflects the degree of correlation. (**D**) Module–trait correlations and corresponding *p*-values. These traits correspond to the eight physiological indexes mentioned above.

**Figure 6 ijms-21-05095-f006:**
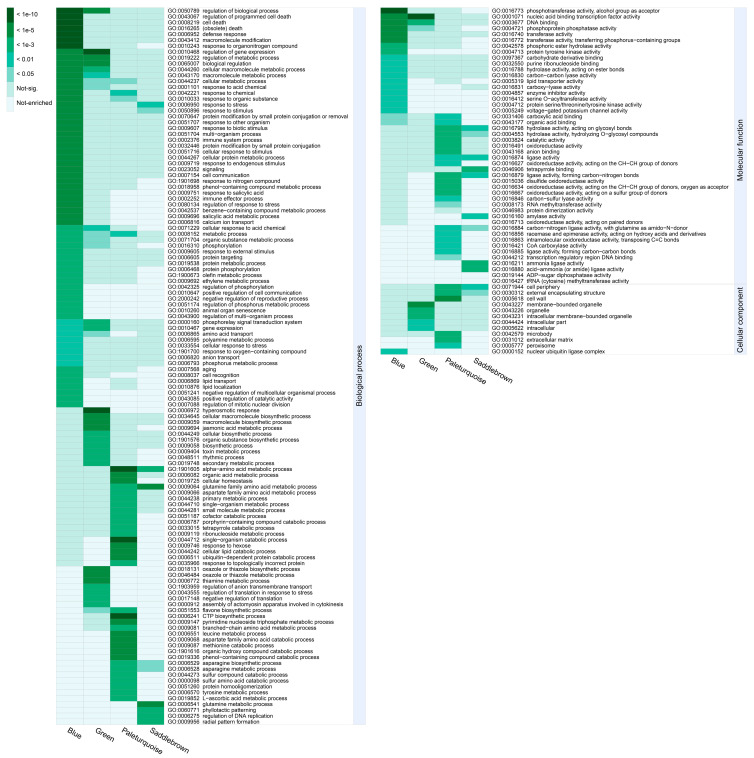
GO enrichment analysis of four significant modules. All significant GO-terms presented in the heatmap have been de-redundant with the REVIGO online platform. Each column represents a significant module, and each row represents a GO-term that is significantly (*p*-value < 0.01) enriched in at least one module. The color scale on the left represents specific intervals of significance level (corrected *p*-value).

**Figure 7 ijms-21-05095-f007:**
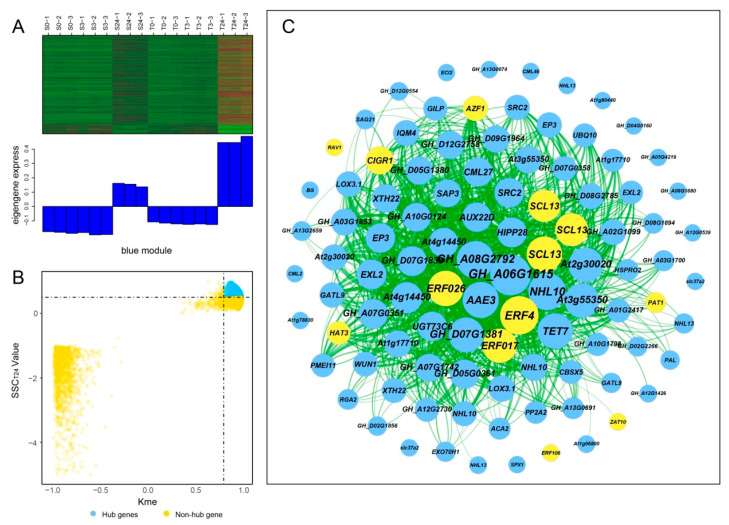
Coexpression network construction of the blue module. (**A**) Gene coexpression heatmap of the blue module (upper panel) and expression level of the corresponding eigengene in each sample (lower panel). (**B**) Dot plot mining hub genes with higher module connectivity and expression levels at T24. The X-axis represents the connectivity values (Kme) of genes in the blue module, while the Y-axis represents the dominant expression value (SSC_T24_) of the genes at T24. Genes with Kme > 0.8 and SSC_T24_ > 0.5 were identified as hub genes (blue dots). (**C**) The coexpression network of hub genes in the blue module. The top 100 hub genes with weight value > 0.45 and higher expression levels at T24 were used to construct the network. The node size indicated gene degree, and the color was used to distinguish whether it was a transcription factor gene (orange node). Line thickness between nodes reflects the weight value.

**Figure 8 ijms-21-05095-f008:**
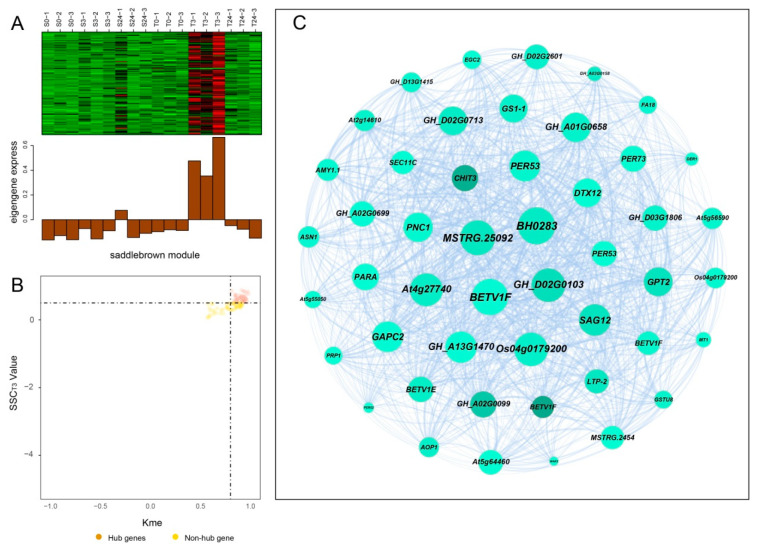
Coexpression network construction of the saddlebrown module. (**A**) Gene coexpression heatmap of the saddlebrown module (upper panel) and expression level of the corresponding eigengene in each sample (lower panel). (**B**) Dot plot mining hub genes with higher module connectivity and expression levels at T3. The X-axis represents the connectivity values (Kme) of genes in the saddlebrown module, while the Y-axis represents the dominant expression value (SSC_T3_) of the genes at T3. Genes with Kme > 0.8 and SSC_T__3_ > 0.50 were identified as hub genes (saddlebrown dots). (**C**) The coexpression network of all coexpressed genes in the saddlebrown module. The node size indicated gene degree. Line thickness between nodes reflects the weight value.

**Figure 9 ijms-21-05095-f009:**
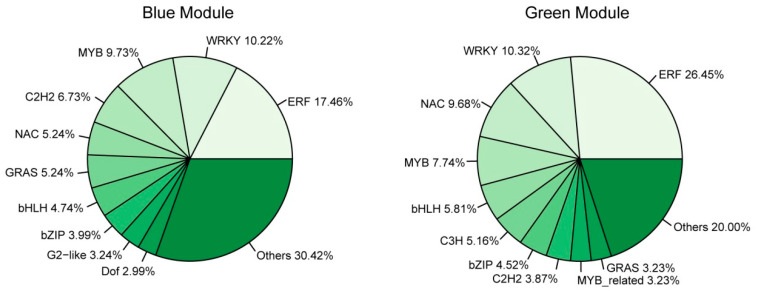
The proportion of genes in the top 10 abundant transcription factor (TF) families in blue (**left**) and green (**right**) modules.

**Figure 10 ijms-21-05095-f010:**
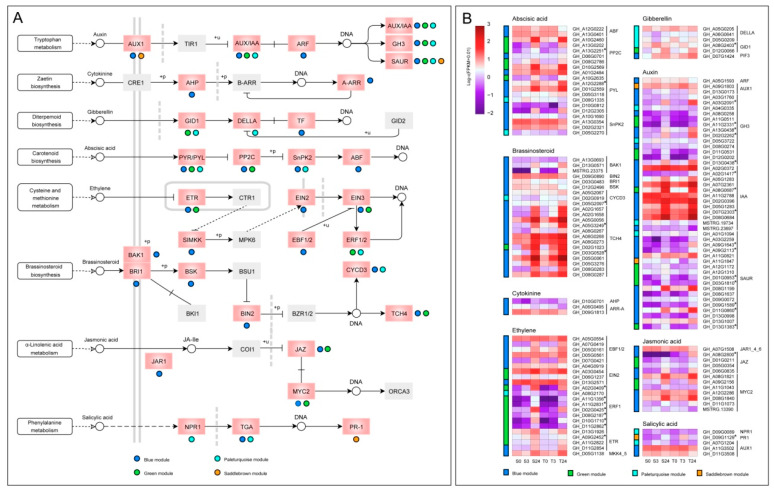
The DEGs of four significant modules in the plant hormone signaling pathway. (**A**) Distribution of genes from four modules on eight plant hormone pathways. The pink boxes denoted the genes enriched, and the colored dots below or to the right of the pink boxes indicated from which module the genes came. The gray boxes denoted the genes that had not been enriched. The meanings of different types of arrows have been strictly defined in KEGG databases (https://www.genome.jp/kegg/document/help_pathway.html). (**B**) The expression patterns of genes distributed on the eight hormone signaling pathways in the four modules. Gene expression data were normalized by log_10_ (FPKM + 0.01). The colored bar on the left of each heatmap represented the modules from which the corresponding genes on the right come from.

**Figure 11 ijms-21-05095-f011:**
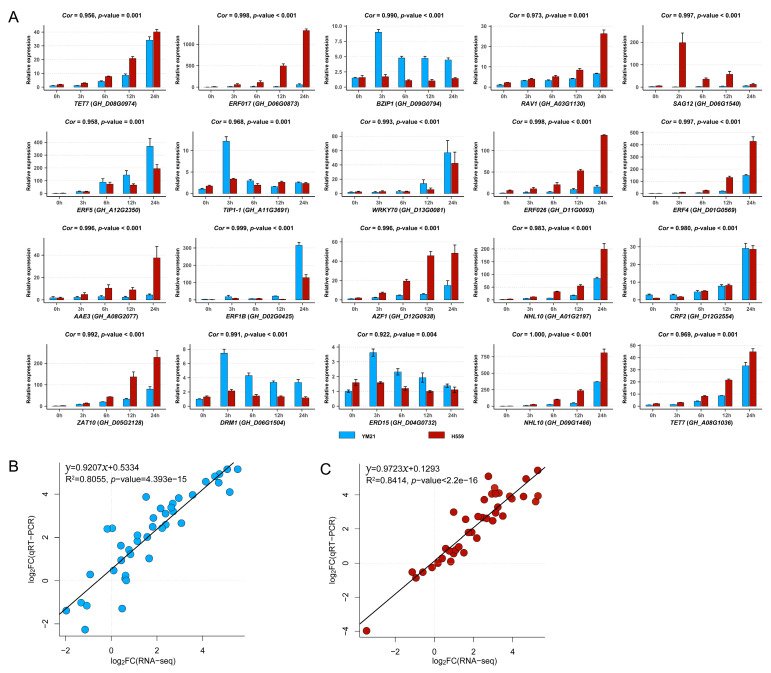
Validation of the expression patterns of 20 genes in four significant WGCNA modules by using qRT-PCR. (**A**) The qRT-PCR results of 20 genes. Here we additionally showed the expression level of each gene at 6 h and 12 h, and only used the qRT-PCR results at 0 h, 3 h, and 6 h under cold stress to perform correlation analysis with RNA-seq data. The correlation coefficient (*Cor*) and correlation significance level (*p*-value) of the expression pattern was added above each histogram. (**B**) Validation of gene expression patterns in cold-sensitive variety YM21. Each point represents a value of fold change of expression level at S3 or S24 compared with that at S0 or S3. (**C**) Validation of gene expression patterns in cold-tolerant variety H559. Each point represents a value of fold change of expression level at T3 or T24 compared with that at T0 or T3. The fold change values were converted by log2 standardization.

**Table 1 ijms-21-05095-t001:** Summary of sequencing data for different samples.

Sample	Raw Data (bp)	Clean Data (bp)	Q30 (%)	GC (%)	Unmapped (%)	Unique_Mapped (%)	Total_Mapped (%)
T0-1	7.66 × 10^9^	7.62 × 10^9^	91.20%	46.09%	3.42%	91.90%	96.58%
T0-2	7.20 × 10^9^	7.17 × 10^9^	92.04%	46.10%	3.24%	92.05%	96.76%
T0-3	7.95 × 10^9^	7.92 × 10^9^	91.27%	45.98%	3.61%	91.62%	96.39%
T3-1	7.59 × 10^9^	7.55 × 10^9^	91.08%	46.40%	3.83%	90.17%	96.17%
T3-2	7.56 × 10^9^	7.52 × 10^9^	92.15%	46.15%	3.19%	91.04%	96.81%
T3-3	7.04 × 10^9^	7.01 × 10^9^	91.68%	46.47%	3.08%	91.00%	96.92%
T24-1	8.19 × 10^9^	8.16 × 10^9^	92.32%	46.28%	3.41%	91.45%	96.59%
T24-2	7.49 × 10^9^	7.46 × 10^9^	91.79%	46.07%	3.08%	91.97%	96.92%
T24-3	8.69 × 10^9^	8.65 × 10^9^	91.75%	46.26%	3.39%	91.31%	96.61%
S0-1	6.53 × 10^9^	6.50 × 10^9^	92.02%	46.17%	3.18%	92.14%	96.82%
S0-2	7.95 × 10^9^	7.92 × 10^9^	91.17%	46.00%	3.51%	91.84%	96.49%
S0-3	8.07 × 10^9^	8.04 × 10^9^	91.45%	45.83%	3.38%	92.01%	96.62%
S3-1	7.05 × 10^9^	7.02 × 10^9^	92.09%	46.78%	2.96%	91.63%	97.04%
S3-2	8.39 × 10^9^	8.36 × 10^9^	91.23%	46.03%	3.90%	90.84%	96.10%
S3-3	8.98 × 10^9^	8.95 × 10^9^	92.33%	46.25%	2.53%	91.83%	97.47%
S24-1	6.89 × 10^9^	6.87 × 10^9^	92.16%	45.93%	2.56%	92.27%	97.44%
S24-2	8.77 × 10_9_	8.73 × 10^9^	91.36%	46.16%	3.39%	91.13%	96.61%
S24-3	8.60 × 10_9_	8.57 × 10^9^	91.55%	46.08%	7.48%	87.60%	92.52%

**Table 2 ijms-21-05095-t002:** The number of differentially expressed genes within and between H559 and YM21 during chilling stress.

Comparison	Total	Upregulated	Downregulated
T0-vs-T3	1626	954	672
T0-vs-T24	13,558	4591	8967
T3-vs-T24	11,902	4224	7678
S0-vs-S3	2030	1056	972
S0-vs-S24	11,795	4522	7273
S3-vs-S24	8984	4152	4832
S0-vs-T0	1504	964	540
S3-vs-T3	2727	1886	841
S24-vs-T24	2316	1397	919

**Table 3 ijms-21-05095-t003:** Significant Kyoto Encyclopedia of Genes and Genomes (KEGG) pathways of four significant WGCNA modules.

Module	Pathway ID	Pathway	Number of Genes	*p*-Value
Blue	ko04626	Plant–pathogen interaction	114	1.60 × 10^−14^
	ko04075	Plant hormone signal transduction	85	1.15 × 10^−9^
	ko04016	MAPK signaling pathway	56	2.58 × 10^−9^
	ko00062	Fatty acid elongation	12	1.66 × 10^−3^
	ko00750	Vitamin B6 metabolism	7	2.65 × 10^−3^
	ko00601	Glycosphingolipid biosynthesis	3	5.61 × 10^−3^
	ko00330	Arginine and proline metabolism	15	9.27 × 10^−3^
Green	ko04016	MAPK signaling pathway	25	1.35 × 10^−8^
	ko04075	Plant hormone signal transduction	34	3.49 × 10^−8^
	ko00730	Thiamine metabolism	6	1.12 × 10^−4^
	ko04712	Circadian rhythm	7	2.02 × 10^−3^
Paleturquoise	ko00071	Fatty acid degradation	8	7.08 × 10^−6^
	ko00592	alpha-Linolenic acid metabolism	7	3.46 × 10^−5^
	ko00280	Valine, leucine, and isoleucine degradation	5	1.03 × 10^−3^
	ko00590	Arachidonic acid metabolism	3	2.58 × 10^−3^
	ko00630	Glyoxylate and dicarboxylate metabolism	6	2.70 × 10^−3^
	ko04075	Plant hormone signal transduction	14	2.74 × 10^−3^
	ko00511	Other glycan degradation	3	3.29 × 10^−3^
	ko00565	Ether lipid metabolism	4	3.86 × 10^−3^
	ko01100	Metabolic pathways	50	9.82 × 10^−3^
	ko00260	Glycine, serine, and threonine metabolism	5	9.92 × 10^−3^
Saddlebrown	ko00940	Phenylpropanoid biosynthesis	6	3.25 × 10^−4^
	ko00480	Glutathione metabolism	4	2.02 × 10^−3^
	ko00250	Alanine, aspartate, and glutamate metabolism	3	2.79 × 10^−3^

**Table 4 ijms-21-05095-t004:** The top five hub genes with high Z-score value in the dominant expression period of each module.

Module (Preponderant Stage)	Gene ID	Symbol	Description	Z-Score	Kme	SSC
Blue (T24)	GH_D12G2915	CML27	probable calcium-binding protein CML27	25.89	0.99	0.54
	GH_D09G1466	NHL10	protein YLS9-like	17.86	0.99	0.55
	GH_A10G2658	WUN1	wound-induced protein 1-like	14.42	0.98	0.64
	GH_D05G2128	ZAT10	zinc finger protein ZAT10-like	13.54	0.98	0.52
	GH_A08G1036	TET7	tetraspanin-8-like	12.91	0.99	0.53
Green (S24)	GH_A13G1006	pcbAB	N-(5-amino-5-carboxypentanoyl)-L-cysteinyl-D-valine synthase	2.75	0.99	0.58
	GH_A08G0687	AUX22D	auxin-induced protein 22D-like	2.41	0.98	0.67
	GH_D12G1406	NIMIN-1	protein NIM1-INTERACTING 1-like	2.39	0.98	0.50
	GH_A05G4197	SIB2	sigma factor binding protein 2, chloroplastic-like	2.13	0.99	0.53
	GH_D02G1771	At1g60420	probable nucleoredoxin 1	1.91	0.90	0.56
Paleturquoise (S3)	GH_D06G1504	DRM1	auxin-repressed 12.5 kDa protein isoform X2	28.36	0.97	0.52
	GH_A10G0258	TSJT1	stem-specific protein TSJT1-like	11.38	0.92	0.55
	GH_A07G0097	OXS3	Oxidative stress 3	10.42	0.94	0.58
	GH_A07G0096	PDD4L	programmed cell death protein 4-like	5.36	0.94	0.55
	GH_A11G3691	TIP1-1	aquaporin TIP1-1-like	2.77	0.97	0.55
Saddlebrown (T3)	GH_D04G1587	BETV1F	pathogenesis-related protein STH-2-like	1.04	0.91	0.67
	GH_A12G2726	CHIT3	hevamine-A-like	0.90	0.94	0.57
	GH_A02G0099	-	uncharacterized protein LOC108475098 [Gossypium arboreum]	0.51	0.90	0.67
	GH_D02G0103	-	uncharacterized protein LOC105796351 [Gossypium raimondii]	0.19	0.93	0.69
	GH_D06G1735	GPT2	glucose-6-phosphate/phosphate translocator 2, chloroplastic-like	0.12	0.90	0.67

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
