# Peer review of "Transcriptomic Profiling of Young Cotyledons Response to Chilling Stress in Two Contrasting Cotton (Gossypium hirsutum L.) Genotypes at the Seedling Stage"

_ijms, 2020, doi:10.3390/ijms21145095_

Round 1

Reviewer 1 Report

Dear author,

I have thoroughly reviewed this manuscript titled “Transcriptomic profiling of young cotyledons response to chilling stress in two contrasting cotton (Gossypium hirsutum L.) genotypes at the seedling stage.

The manuscript is worthy of being published in this high esteem journal due to the vital information embedded in it.

However, figure 4 and 6 seems a blur. It is highly recommended to change it.

Overall, I recommend this article as a “minor revision” and submit again for further consideration.

Author Response

Point 1: However, figure 4 and 6 seems a blur. It is highly recommended to change it.

Response 1: For Figure 4, we reset the color scale of the bubbles and set a clear edge for each bubble. To make the text in the figure easy to view, we referred to the figure styles in the two references (Liang et al., 2015, DOI: 10.1371/journal.pone.0143503; Polonio et al., 2019,  DOI: 10.1186/s12864-019-5938-0), and put the text into a black rectangle box and drew it out from the corresponding bubble. We think the revised figure is clearer, and then replace the Figure 4 in the original manuscript with the revised version. For Figure 6, we changed the length and width of each box in the heatmap, replaced the original gray font with black, and eliminated unnecessary text background. The original figure is a bit blurry even if it is enlarged, but the revised figure will not have this problem again.

Reviewer 2 Report

The manuscript of Cheng et al. present extensive results obtained through a RNAseq transcriptomic approach concerning the mechanisms of resistance to chilling stress in cotton cotyledons.

As a first step, the authors selected among 74 cotton varieties the two most contrasted ones in term of chilling sensitivity of seedlings at the cotyledon development stage. The selection of these two varieties is pretty convincing and well presented. These two contrastred varieties were then used to compare physiological status and RNAseq transcriptomic profiles in control and chilling stress conditions. The data obtained are of excellent quality. These raw data were then used in several bioinformatics analysis in order to obtain informations concerning genes differential expression analysis according to varieties and response to stress, gene co-expression networks, GO and KEGG pathway annotations, identification of genes being important crossroads in this system. Author’s interest concerned more particularly genes coding for transcription factors and involved in plant hormones transduction pathways. Finally, the RNAseq data were validated by RT-qPCR on 20 cotton genes.

The manuscript is clearly presented, the text quality is excellent and the conclusions discussed in the final part of the manuscript are extensive and well documented. Consequently, the data presented in this manuscript are useful for the plant research community interested in crop resistance to chilling stress and deserve publication in Int. J. Mol. Sci.

However, I recommend the authors some modifications of the text which could, according to me, improve its quality:

  • The authors should be careful with the terms they use in their interpretation of the data. As an example, on line 221, the authors wrote “These results fully demonstrated that…”. According to me, the data presented “suggest that…” but can’t lead to a full demonstration on what the authors claim. This problem is mainly encountered in the “results” part of the manuscript and the authors are much more careful on the terms used in the “discussion” part.
  • The qPCR validation data presentation should be changed. First, in figure 11, the name of the genes corresponds to their genomic code. This must be replaced by the putative name of the gene (indicating their putative function) as listed in table S5. More importantly, the authors need to indicate the criteria they used when they selected these genes for RT-qPCR validation. Where they selected because of a specific interest, randomly or because of their expression profile? The criteria of selection could create a bias in the interpretation of the results, especially according to validation of a full set of transcriptomic data. A last question to be answered by the authors concerning the RT-qPCR data concern the choice of the reference gene (GH Actin). The authors need to indicate, at least, a bibliographical reference indicating the full validation of this reference gene in cotton for RT-qPCR experiments.

Minor modifications:

-line 206 : “oof” needs to be changed to “of”

-line 241 : “paleurquoise” needs to be changed to “paleturquoise”.

Author Response

Point 1: The authors should be careful with the terms they use in their interpretation of the data. As an example, on line 221, the authors wrote “These results fully demonstrated that…”. According to me, the data presented “suggest that…” but can’t lead to a full demonstration on what the authors claim. This problem is mainly encountered in the “results” part of the manuscript and the authors are much more careful on the terms used in the “discussion” part.

Response 1: On line 221, we have changed "fully demonstrated that..." to "suggest that..." in accordance with your wise comments. To ensure that similar problems do not occur, we have reviewed the "results" section and found no similar exaggerations.

Point 2: The qPCR validation data presentation should be changed. First, in figure 11, the name of the genes corresponds to their genomic code. This must be replaced by the putative name of the gene (indicating their putative function) as listed in table S5. More importantly, the authors need to indicate the criteria they used when they selected these genes for RT-qPCR validation. Where they selected because of a specific interest, randomly or because of their expression profile? The criteria of selection could create a bias in the interpretation of the results, especially according to validation of a full set of transcriptomic data. A last question to be answered by the authors concerning the RT-qPCR data concern the choice of the reference gene (GH Actin). The authors need to indicate, at least, a bibliographical reference indicating the full validation of this reference gene in cotton for RT-qPCR experiments.

Response 2: (1) According to the requirements, we have added the putative name of each gene in Figure 11A and presented it with the genomic code together in the figure. In addition, we have added the putative functional description of each gene to Table S5. However, when we tried to replace the genomic code with the putative name of each gene, we found that NH10 and TET7 respectively corresponded to two genes with different genomic codes, so we had to use the form of "gene name (genomic code)" to avoid misunderstanding. (2) Based on your concerns about the criteria for selection of genes used for qRT-PCR validation, we will give you a detailed explanation and supplement important information in the corresponding part of the results. First, an important purpose of our transcriptome project is to obtain interesting hub genes, and to provide help for further study of gene function and explore the molecular mechanism of cold resistance of cotton seedlings. These twenty genes are some interesting hub genes in the four significant WGCNA modules, some of which are specifically mentioned in the manuscript, such as NHL10, RAV1, TET7, TIP1-1, ERF5, ERF4, ERF1B, ZAT10, ERF017, DRM1, BZIP1 and ERF026. According to your request, we have made a supplement in the corresponding result section. For example, on line 410 we add "and obtain candidate genes for further functional studies" after "RNA-seq data", and expand "twenty genes" to "twenty interesting hub genes" on line 411. (3) For the GhACTIN7 gene, we have added a reference (reference 100) in the corresponding position of the "method" section (on line 637). This reference is about a DTX/MATE gene from cotton to enhance the abiotic stress resistance in transgenic Arabidopsis. The internal reference gene used in our qRT-PCR validation is the same as that used in the reference. In another document we found the NCBI accession number (AY305733) of the gene. We used the AY305733 number to search for the nucleic acid sequence in NCBI, and then found the genomic code of the gene on the CottonFGD website and added it to Table S5.

Point 3: Minor modifications: (1) line 206 : “oof” needs to be changed to “of”. (2) line 241 : “paleurquoise” needs to be changed to “paleturquoise”.

Response 3: (1) On line 206, "oof" has been corrected  to "of" as requested. (2) On line 241, "paleurquoise" has been corrected  to "paleturquoise" as requested.